# Correlates of Exercise Behavior Based on Socio-Ecological Theoretical Model among Chinese Urban Adults: An Empirical Study

**DOI:** 10.3390/bs14090831

**Published:** 2024-09-17

**Authors:** Yong Zhang, Ya-Jun Zhang, Yongdong Qian, Zhaofeng Meng, Xiaofang Ying

**Affiliations:** 1School of Medicine, Shaoxing University, Shaoxing 312000, Chinayajunzhang@usx.edu.cn (Y.-J.Z.); 2College of Physical Education and Health Sciences, Zhejiang Normal University, Jinhua 321004, China; tyxyqyd@zjnu.cn; 3School of Business, Shaoxing University, Shaoxing 312000, China

**Keywords:** Chinese urban adults, exercise behavior, correlates, socio-ecology

## Abstract

Background: Identifying the correlates of exercise behavior is essential to combating physical inactivity as a public health concern. The purpose of this study is to identify the correlates of physical activity among urban Chinese adults under the social-ecological theoretical model in order to facilitate targeted interventions to promote physical activity. Methods: Using the socio-ecological model, we conducted a questionnaire survey among 1459 urban residents in Zhejiang and Shaanxi provinces of China, collecting data on individual demographic factors, sociological factors, environmental perception, and exercise behavior. Binary logistic regression models were employed to analyze the relationships between exercise behavior and socio-ecological factors. Results: Male gender (*p* < 0.01), advanced age (*p* < 0.001), higher education level (*p* < 0.05), living independently from parents (*p* < 0.05), absence of childcare responsibilities (*p* < 0.01), residence in a county/prefecture-level city (*p* < 0.001), favorable neighborhood esthetics (*p* < 0.001), available greenways/parks (*p* < 0.001), and family support for exercise participation (*p* < 0.05) were significantly correlated with an increased likelihood of participating in physical activity. Male gender (*p* < 0.001), advanced age (*p* < 0.001), absence of childcare responsibilities (*p* < 0.05), good neighborhood vegetation (*p* < 0.01), availability of free neighborhood exercise facilities (*p* < 0.05), and support from friends for exercise participation (*p* < 0.01) were significantly correlated with an increased likelihood of engaging in physical activity for more than 150 min per week. BMI, community air quality, traffic safety, public safety, and level of social development were not major correlates. Conclusions: To promote exercise behavior, more attention should be paid to individuals who are female, young, have lower levels of education, bear childcare responsibilities, or reside in provincial capitals in China. Improving the habitat environment and providing convenient and affordable facilities should also be considered. Furthermore, support from family and friends can positively reinforce exercise behavior.

## 1. Introduction

With the development of the economy, society, and urbanization, public health problems related to physical inactivity have become a major social concern in China. Physical exercise behavior plays a positive role in promoting physical activity. However, the effectiveness of promoting physical activity behavior among adults is suboptimal due to the multidimensional and complex nature of factors influencing their exercise participation as a result of high levels of socialization [1]. With the implementation of the “National Fitness Program” [2] and “Healthy China 2030” [3], improving adult exercise behavior has become an imperative issue for the government to address. Therefore, it is of great practical significance to identify the constraining components of exercise behavior among Chinese urban residents in order to select effective intervention strategies to promote physical activity.

Current perspectives on exercise behavior intervention are based primarily in pedagogy, cognitive science, psychology, and behavioral science. The mainstream theories have successively included the Health Belief Model [4], the Trans-theoretical Model [5], the Expectancy Value Model [6], the Planned Behavior Theory [7], and the Self-Determination Theory [8]. However, research based on these theories has not yielded the expected results. The main argument is that despite the observed positive effects of health beliefs, planned behavior, and self-determination on exercise behavior, they do not necessarily translate into actual exercise engagement [9]. It is widely accepted that exercise behavior is the result of the comprehensive effects of individual and environmental factors [1], and therefore studies that focus on certain isolated factors often produce biased results because they fail to account for the influence of other factors or their interactions with the dependent variables. From a behavioral perspective, individual behavior is determined by multiple factors that interact in complex ways, sometimes superimposing and reinforcing each other, sometimes offsetting and attenuating each other. Therefore, it is necessary to conduct multilevel and multidimensional empirical studies on the correlates of adult exercise behavior within a socio-ecological framework [10,11,12].

The socio-ecological theoretical model emphasizes that behavior is not solely the result of individual choices but is shaped by a complex interplay of factors at multiple levels, ranging from personal characteristics to broad environmental and societal influences [11]. The comprehensive systems study based on the socio-ecological framework has a methodological advantage that overcomes the outcome bias of the fragmented study of other theories [11]. From previous studies, factors related to exercise behavior based on the socio-ecological model include four levels: individual, interpersonal, environmental, and policy, and empirical research mainly focused on the individual, interpersonal, and environmental levels [10,11,13,14]. Among them, the individual level focuses on socio-demographic characteristics, the interpersonal level focuses on family and peer support, and the environmental level focuses on neighborhood environmental factors (including the natural, built, and social environments) that are closely related to daily life [10,15,16]. Although some studies have been conducted in European and American countries using a socio-ecological approach to address factors that influence physical activity [13,17,18,19], the results are not necessarily applicable to China due to differences in socio-economic and cultural backgrounds. Currently, there are few studies based on social ecology theory to determine the influencing factors of exercise behavior in China, with only two studies specifically targeting the elderly [20,21]. Further research on adults still needs to be conducted to complement existing studies, which has practical implications for promoting exercise behavior intervention strategies in China.

Exercising for at least 150 min per week is recommended by the World Health Organization [22]. Exercise participation and total exercise time per week are two key variables representing exercise behavior with progressive relationship. The objective of this study is to employ a comprehensive socio-ecological model to identify and analyze the main correlates of exercise participation and total exercise time per week among current urban residents in China, with the aim of facilitating targeted interventions for promoting their exercise behavior.

## 2. Materials and Methods

### 2.1. Study Design and Data Collection

This cross-sectional study was conducted using stratified sampling among urban adults at different levels of social development in Zhejiang Province (a developed province) and Shaanxi Province (a developing province) in China. Data were collected between April and June 2021 using an online questionnaire. The questionnaire consisted of three sections. The first section focused on the socio-demographic characteristics of individuals, including gender, age, height, weight, marital status, employment status, education level, household income, urban scale and family care, and local development level. The second section focused on the socio-ecological variables that influence exercise behavior, including the natural environment, built environment, social environment, exercise facilities, and interpersonal support. The third section focused on the characteristics of exercise behavior, including exercise participation and total exercise time per week. Interpersonal and facility variables were assessed with “yes” or “no”. Environment variable scores were assigned using the Likert 5-subscale method, where each variable is assigned 1, 2, 3, 4, and 5 points, from very poor to very good. Recent exercise participation behavior was rated as “yes” or “no”, and total exercise time per week was divided into <150 min and ≥150 min. The study was approved by the ethics committee of Zhejiang Normal University (ZSRT2021014). The respondents of the questionnaire remain completely anonymous, and they were fully informed about the purpose and content of the study before choosing to participate voluntarily.

### 2.2. Sample Size

A total of 1459 valid survey responses were included in the statistical analysis. The sample size was analyzed using G * Power 3 software according to the conventional standard (OR = 1.3) and significance level standard (alpha = 0.05), resulting in a study sample power of 0.979.

### 2.3. Validity and Reliability of the Questionnaire

The process of constructing the survey questionnaire included the following: (1) a review of existing research literature; (2) focus group meetings and targeted urban adults; (3) a panel of experts; (4) confirmation of the clarity of item wording; and (5) data analysis.

The literature review included a synthesis of constructs derived from socio-ecological studies and measures that involved assessments of individual, interpersonal, and environmental resources in exercise behavior [1,10,11,13,14,20]. Focus group meetings were used to discuss topics and issues related to individual, interpersonal, and environmental resources in adult exercise behaviors. These preliminary efforts resulted in the generation of an initial pool of survey items designed to assess exercise behavior resources. Then, a panel of 10 experts in the areas of public health, exercise and health promotion, exercise behavior research, and sports psychology was used to further review and scrutinize the content of these items employing a Delphi technique. The responses from these panel members were examined, and a decision was then made on whether to modify, delete, or add items based on the 80% criteria. Subsequently, interviews were conducted with 30 adults to examine if the questionnaire content was expressed clearly and to modify any unclear wording. Finally, factor analysis was performed on the socio-ecological content other than the demographic variables in the questionnaire, using 500 pre-survey samples. According to conventional criteria, components with values below 0.4 were excluded [23], and three factors were derived, namely environment, facilities, and interpersonal support. The KMO value was 0.728 (≥0.7), and the cumulative contribution of variance interpretation was 71.91%, indicating a good structural validity of the questionnaire [24]. In addition, the loadings of relevant components of each factor were good (0.679–0.862) based on the rotation component matrix. Reliability tests for each factor indicated that the Cronbach’s α values were 0.831, 0.812, and 0.806 for environment (esthetics, vegetation, air quality, public safety, and traffic safety), facilities (greenway/park, free facilities, and paid facilities), and interpersonal support (family support, friend support, and physician advice), respectively. Cronbach’s alpha was 0.844 for all factors. Overall, the internal consistency and reliability of the questionnaire were good [25].

### 2.4. Data Analysis

All data were processed using SPSS 21.0, and the significance level was set at *p* < 0.05. Binary logistic regression models were used to analyze the relationships between exercise participation (participation and non-participation), total exercise time per week (<150 min, ≥150 min), and related socio-ecological factors. The dependent variables were exercise participation and total exercise time per week, and the independent variables were the socio-demographic characteristics and the socio-ecological factors of the individual. Analyses were conducted using the backward stepwise method (likelihood ratio), with covariates removed according to the conventional standard (alpha < 0.10).

## 3. Results

### 3.1. Socio-Demographic Characteristics of Participants

All the socio-demographic variables of the participants are shown in Table 1. The socio-demographic characteristics of the sample showed a relatively reasonable distribution of gender, age, body mass index (BMI), region, marital status, employment status, and social status (education level, household income).

### 3.2. Socio-Ecological Factors of Individual Exercise Behavior

Table 2 shows the results for neighborhood environment, facilities, and interpersonal support. Neighborhood esthetics, air quality, and vegetation were all between “average” and “good”. Neighborhood crime safety was close to “good” both day and night, but traffic safety was not good enough. Most respondents reported that there were greenways/parks or free facilities available for exercise in their neighborhoods. In addition, more than 90 percent of the respondents reported family or friend support for their exercise behavior. Nearly 70 percent of the respondents reported physician advice to exercise.

### 3.3. Characteristics of Exercise Behavior

A fraction of respondents (74.30%) reported that they had recently participated in physical activity, with only 52.86% of this population exceeding 150 min per week.

### 3.4. Correlates of Exercise Participation

The variables retained in the exercise participation model were gender, age, employment status, education level, living with parents, childcare, urban scale, esthetics, greenways/parks, and family support (Table 3). The variables not included in the model were BMI, marital status, income, elderly care, social development level, air quality, vegetation, free facilities, paid facilities, day security, night security, traffic safety, friend support, and physician advice. The prediction accuracy of the model was 75.68%, the model was valid (*p* < 0.001), and the goodness of fit was good (*p* = 0.797).

According to the model results, we observed that the following factors significantly increased the likelihood of exercise participation: male gender (*p* < 0.01), advanced age (*p* < 0.001), higher education level (*p* < 0.05), independent living status (*p* < 0.05), absence of childcare responsibilities (*p* < 0.01), residence in county-level or prefecture-level cities (*p* < 0.001), favorable neighborhood esthetics (*p* < 0.001), available greenways/parks (*p* < 0.001), and family support for exercise (*p* < 0.05).

### 3.5. Correlates of Total Exercise Time Per Week

The results show that gender, age, living with parents, childcare, vegetation, free facilities, friend support, and physician advice were in the total exercise time model (Table 4). The prediction accuracy of the model was 60.89%, the model was valid (*p* < 0.001), and the goodness of fit was good (*p* = 0.117). In the process of model analysis, the variables of BMI, marital status, employment status, education level, income, elderly care, social development level, urban scale, esthetics, air quality, greenways/parks, paid facilities, day security, night security, traffic safety, and family support were excluded.

From the model, we found that the following factors significantly increased the likelihood of exercise participation for more than 150 min per week: male gender (*p* < 0.001), advanced age (*p* < 0.001), absence of childcare responsibilities (*p* < 0.05), good neighborhood vegetation (*p* < 0.001), free exercise facilities in the neighborhood (*p* < 0.05), friend support for exercise participation (*p* < 0.001), and lack of physician advice for exercise participation (*p* < 0.05).

## 4. Discussions

Participation and level of physical activity are the primary components used to assess exercise behavior and determine whether an individual is exercising sufficiently or not. The purpose of this study was to examine the personal, social, and environmental correlates of exercise behavior among Chinese urban adults. As far as we know, this is the first study on the exercise participation and amount of Chinese urban residents based on the socio-ecological model. The study revealed that the primary barriers to participation in physical activity are associated with demographic factors such as being female, being young, having a lower education level, living with parents, having childcare responsibilities, residing in provincial capital cities or megacities, lacking good neighborhood esthetics and available greenways/parks, and lacking family support for exercise (Table 3). The study also found that being female, being young, having childcare responsibilities, not having good neighborhood vegetation, not having free exercise facilities in the neighborhood, not having friend support for exercise, and not having physician advice for exercise were unfavorable factors for meeting the exercise guidelines (Table 4). In addition, the study found that factors such as BMI, marital status, income, elderly care, social development level, air quality, paid facilities, public security, and traffic safety did not significantly correlate with participation and level of physical activity among Chinese urban adults (Table 3 and Table 4).

From the perspective of individual demographic characteristics, the results of this study regarding gender, age, and childcare responsibilities were consistent with previous studies [16,17,26,27,28,29], and adequate time and energy may still be important for individual exercise behavior [16,29,30]. Family caregiving may consume young women’s time and energy, affecting their exercise behaviors. It is also worth noting that living with parents may be detrimental to the exercise behavior of urban adults in China, and we hypothesize that living with parents may somewhat constrain individual behavioral activities. In addition, some differences from previous studies were also found in this study. Although some studies suggested that being overweight or obese was a detrimental factor for exercise [27,31], our research did not find any evidence to support this claim. Previous studies also supported a positive association between education level and income and exercise behavior [20,28], which was thought to be related to the greater availability of leisure time among high-income people [32]. We also believe that people with higher incomes tend to have higher levels of education, which in turn is associated with a better understanding of the benefits of exercise and increased self-efficacy for physical activity. However, in this study, we did not find that income level was the correlate of exercise behavior among Chinese urban residents. The study also found that there was no significant association between exercise behavior and marital status, which is consistent with previous research findings [14,27].

In this study, for the first time, the level of social development and urban scale were included in the analysis model. It has been generally believed that there is a positive correlation between exercise behavior and the level of social development; the more advanced the level of social development, the more likely it is that one will have a better exercise habit. However, the study showed that there was no obvious correlation between the components of exercise behavior and the level of social development. Although big cities may offer advantages in terms of policy, guidance, publicity, and sports events, these distal factors do not have a significant positive effect on individual exercise behavior. It was observed that the exercise behavior of residents in provincial capitals was not as good as that of residents in small and medium-sized cities. This is because proximal factors closely related to an individual’s daily life, such as energy, time, neighborhood environment, and exercise facilities, may serve as the key determinants of their exercise behavior, and small or medium-sized cities in China typically have advantages in such proximal factors.

Previous studies have also found that the neighborhood environment, such as esthetics, traffic, and safety, is associated with levels of physical activity participation [13,29,33]. Among them, the positive effect of esthetics on physical activity behavior was confirmed [13,33,34,35], with neighborhood vegetation coverage being a key factor in promoting environmental esthetics [36]. Our research results also indicated that the neighborhood environment should be considered in urban planning and development. However, it was worth noting that air quality was no longer the primary determinant of Chinese urban residents’ exercise behavior. Some studies also suggested a positive association between neighborhood safety and exercise behavior [29,37,38], suggesting that poor perceptions of environmental safety may hinder outdoor exercise. However, our study found no significant association between neighborhood traffic and safety and exercise behavior among Chinese urban residents, which is consistent with the results of studies in Brazil [39,40]. We believe that differences in neighborhood safety and individual perceptions of self-safety may account for the different findings in these studies. Better traffic safety, less crime, and better perceptions of safety among residents in China may mitigate their negative effects on physical activity behavior. It may also be related to the fact that Chinese urban residents prefer to exercise in their neighborhoods. A study of elderly Chinese citizens found that traffic and safety were not significant factors influencing their leisure-time physical activities because these activities mainly took place in the squares and parks near their communities, which were usually safe [41].

Exercise facilities are closely related to exercise behavior, and the availability and convenience of facilities play a positive role in exercise behavior [26,29,42,43]. Some previous studies in China also confirmed that the accessibility and convenience of exercise facilities had a significant effect on residents’ exercise participation [15,20,44,45]. This study also found evidence that convenient and free exercise facilities, especially greenways/parks near residential areas, played a positive role in promoting physical activity participation.

Social support and interpersonal networks play an important role in the initiation and maintenance of exercise behavior, which is not only consistent with the social–ecological model and Bandura’s social–cognitive theory but also supported by previous studies conducted in Europe and the United States [46,47], South Korea [48], Japan [49], and China [20,45,50]. Exercise support from family and friends directly contributes to physical activity participation, and a good exercise social network can also influence exercise behavior by sharing resources, norms, and attitudes. According to the homogeneity principle, individuals prefer to associate with people with similar characteristics. Exercise participants tend to socialize more with other exercise participants, which has a positive effect on the maintenance of exercise behavior. The results of this study also supported the positive effect of social support from family and friends on exercise initiation and maintenance, but no evidence was found to support the effectiveness of physician advice in promoting exercise behavior. To our knowledge, few studies of the socio-ecological correlates of exercise behavior have included physician advice. Although Andersen’s study included physician support, they did not conduct a separate analysis in their research process (the total score was calculated together with family and friend support as interpersonal support in the study) [27]. Surprisingly, our study found that individuals who received exercise advice from a physician tended to have lower levels of physical activity than those who did not. As a cross-sectional study, we believed that such results might be related to the better health status of those with good exercise behavior and the lower likelihood of receiving exercise advice from physicians, and it also showed that exercise advice from physicians did not play a significant role in promoting exercise behavior among the inactive. In fact, previous studies had found that the effect of exercise advice from physicians on improving exercise behavior was limited [51,52]. Although some physicians provided exercise prescription and counseling to their patients, only 12–42% of patients completed the recommended exercise prescription program [51]. Physician exercise counseling had no significant effect on improving exercise behavior, and implementation of exercise prescription still needs effective interventions from other resources [52].

These findings may have important implications for the promotion of physical activity. Individuals’ exercise behavior is related to their demographic characteristics and socio-ecological factors. Socio-cultural and socio-economic differences between countries may limit the generalizability of previous findings on the correlates of physical activity. This was the first comprehensive study on the correlates of exercise behavior (including participation and level of physical activity) among urban Chinese adults based on the socio-ecological framework. Therefore, the current findings may be useful in developing interventions to promote exercise behavior among Chinese urban residents.

We also acknowledge several limitations of this study. First, the socio-ecological theory involves many factors, and the research based on the socio-ecological theoretical model involved so many variables that it was difficult to discuss them all in depth and detail in a single study. Second, due to the nature of the cross-sectional design, this study cannot establish a causal relationship between exercise behavior characteristics and demographic and socio-ecological variables. Third, the large number of independent variables used in this study posed a challenge to stratified sampling. Fortunately, the demographic and socio-ecological characteristics of the sample in this study were relatively well distributed. In the future, longitudinal studies can be strengthened based on existing research findings to explore the causal relationship between factors and exercise behavior.

## 5. Conclusions

Currently, about three-quarters of Chinese urban residents participate in physical activity, but their weekly exercise time needs to be improved. To promote their exercise behavior, more attention should be paid to people who are female, young, have lower education levels, have childcare responsibilities, or live in provincial capitals. Improving the habitat environment and providing convenient and affordable exercise facilities should also be considered. In addition, social support from family and friends may enhance exercise participation, but physician advice to exercise may not have the same effect. It is worth noting that individual weight, air quality, traffic safety, public safety, income, and social development level are not the correlates of exercise behavior among Chinese urban residents.

## Figures and Tables

**Table 1 behavsci-14-00831-t001:** Socio-demographic characteristics of the study sample.

Variable	N (%)	Variable	N (%)
Gender		Household income (¥/year)	
Male	597 (40.92)	<30,000	151 (10.35)
Female	862 (59.08)	30,000–50,000	228 (15.63)
Age (years)		50,001–100,000	335 (22.96)
18–34	597 (40.92)	100,001–150,000	286 (19.60)
35–44	470 (32.21)	150,001–300,000	336 (23.03)
45–54	314 (21.52)	>300,000	123 (8.43)
55–	78 (5.35)	Living with parents	
BMI (kg/m^2^)		Yes	522 (35.78)
Obese	80 (5.48)	No	937 (64.22)
Overweight	385 (26.39)	Elderly care	
Underweight	84 (5.76)	Yes	897 (61.48)
Normal weight	910 (62.37)	No	562 (38.52)
Marital status		Child care	
Married	1090 (74.71)	Yes	918 (62.92)
Single	369 (25.29)	No	541 (37.08)
Employment status		Urban scale	
Full-time	1185 (81.22)	Provincial capital	297 (20.36)
Part-time	71 (4.87)	Prefecture-level	556 (38.11)
Unemployed	203 (13.91)	County-level	606 (41.54)
Education level		Local development level	
<high school	197 (13.50)	Developed	719 (49.28)
High school	193 (13.23)	Underdeveloped	740 (50.72)
College/university	840 (57.57)		
Graduate	229 (15.70)		

**Table 2 behavsci-14-00831-t002:** Exercise environment, facilities, and interpersonal support in the neighborhood.

Interpersonal	N (%)	Facilities	N (%)	Environment	Mean (SD)
Family support		Greenways/Parks		Aesthetics	3.61 (0.91)
No	95 (6.51)	No	284 (19.47)	Air quality	3.39 (0.95)
Yes	1364 (93.49)	Yes	1175 (80.53)	Vegetation	3.63 (0.87)
Friend support		Free facilities		Day security	4.05 (0.80)
No	88 (6.03)	No	304 (20.84)	Night security	3.93 (0.79)
Yes	1371 (93.97)	Yes	1155 (79.16)	Traffic safety	2.82 (0.95)
Physician advice		Paid facilities			
No	455 (31.19)	No	421 (28.86)		
Yes	1004 (68.81)	Yes	1038 (71.14)		

**Table 3 behavsci-14-00831-t003:** Correlates of exercise participation.

Variable	B	S.E.	Sig.	Exp (B)	Exp (B) 95% CI	
					Lower	Upper
Gender -male	0.46	0.14	0.001	1.58	1.21	2.07
-female						
Age (years)			0.000			
18–34	−1.18	0.38	0.002	0.31	0.15	0.65
35–44	−0.71	0.39	0.067	0.49	0.23	1.05
45–54	−0.06	0.40	0.879	0.94	0.43	2.07
55–						
Employment status			0.010			
Full-time	−0.39	0.22	0.070	0.68	0.44	1.03
Part-time	0.55	0.39	0.158	1.74	0.81	3.73
Unemployed						
Education level			0.023			
<High school	−0.78	0.27	0.004	0.46	0.27	0.78
High school	−0.62	0.26	0.018	0.54	0.32	0.90
College/university	−0.35	0.20	0.078	0.71	0.48	1.04
Graduate						
Living with parents -No	0.31	0.14	0.023	1.36	1.04	1.78
-Yes						
Childcare -No	0.50	0.15	0.001	1.65	1.23	2.23
-Yes						
Urban scale			0.000			
Provincial capital	−0.75	0.18	0.000	0.47	0.33	0.67
Prefecture-level	−0.24	0.15	0.109	0.78	0.58	1.06
County-level						
Aesthetics	0.38	0.08	0.000	1.47	1.26	1.71
Greenways/Parks -No	−0.63	0.16	0.000	0.53	0.39	0.73
-Yes						
Family support -No	−0.47	0.24	0.049	0.63	0.40	1.00
-Yes						
Constant	1.07	0.53	0.045	2.92		

Model χ^2^ (df) = 176.129 (16), *p* < 0.001, R^2^ = 0.167 (Nagelkerke). Goodness of fit statistics: χ^2^ (df) = 4.624 (8), *p* = 0.797. Gender, age, BMI, marital status, employment status, education level, household income, living with parents, elderly care, childcare, urban scale, local development level, esthetics, air quality, vegetation, greenways/parks, free facilities, paid facilities, day security, night security, traffic safety, family support, friend support, and physician advice were input as independent variables in step 1.

**Table 4 behavsci-14-00831-t004:** Correlates of total exercise time in a week.

Variable	B	S.E.	Sig.	Exp (B)	Exp (B) 95% CI	
					Lower	Upper
Gender -male	0.47	0.13	0.000	1.61	1.25	2.07
-female						
Age (years)			0.000			
18–34	−0.92	0.30	0.002	0.40	0.22	0.71
35–44	−0.82	0.30	0.005	0.44	0.25	0.78
45–54	−0.08	0.30	0.799	0.93	0.52	1.67
55–						
Living with parents -No	0.23	0.14	0.096	1.26	0.96	1.67
-Yes						
Childcare -No	0.33	0.15	0.027	1.39	1.04	1.87
-Yes						
Vegetation	0.26	0.08	0.001	1.30	1.11	1.52
Free facilities -No	−0.34	0.17	0.048	0.71	0.51	1.00
-Yes						
Friend support -No	−1.09	0.32	0.001	0.34	0.18	0.63
-Yes						
Physician advice -No	0.29	0.14	0.045	1.33	1.01	1.76
-Yes						
Constant	−0.70	0.42	0.096	0.49		

Model χ^2^ (df) = 91.708 (10), *p* < 0.001, R^2^ = 0.108 (Nagelkerke). Goodness of fit statistics: χ^2^ (df) = 12.857 (8), *p* = 0.117. The independent variables entered in step 1 were the same as those in Table 3.

## Data Availability

All the data for this study will be made available upon reasonable request.

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
