# Peer review of "Correlates of Exercise Behavior Based on Socio-Ecological Theoretical Model among Chinese Urban Adults: An Empirical Study"

_behavsci, 2024, doi:10.3390/bs14090831_

Round 1

Reviewer 1 Report

Comments and Suggestions for Authors

General evaluation

This study investigates the personal, social, and environmental correlates of exercise behavior among Chinese urban adults under a socio-ecological model. It is a highly relevant and current topic of interest for health professionals and society in general. 

The title provides information about the problem under investigation. 

The abstract states the background and the study’s purpose, basic procedures (selection of study participants, settings, measurements, and analytical methods), main findings and principal conclusions.

As a strength, this article used a large sample (1,459 urban Chinese residents), and about 30% of the cited references are from the last 5 years. A weak point of this article is the lack of information on the ecological model, which is needed for a better understanding the complexity of human behavior.

Introductuion

The introduction should provide a detailed analysis and presentation of the Socio-Ecological Model, as well as highlight what differentiates this model from others. To interpret behavior using the Socio-Ecological Model, it is essential to understand how different levels of influence interact and shape individual and collective actions. This model highlights that behavior is not solely the result of individual choices but is also shaped by a complex interplay of factors at multiple levels, ranging from personal characteristics to broad societal influences.

Methods 

The methods, tools and procedures are well described.

Results

All the results are presented in tables that display the data appropriately, providing a very good interpretation of the information.

Discussion

The results are discussed and compared with other investigations. However, given the use of such a powerful and complex model, a more comprehensive discussion would be expected. To interpret behavior using the Socio-Ecological Model, it is essential to analyze how these levels interact to influence behavior and to develop effective strategies for promoting an active and healthy lifestyle.

Conclusions

The results address the objectives.

The authors state the limitations of the study  

The implications of the results are established the authors could specify some suggestions for future research given the problems encountered.

Reviewer 2 Report

Comments and Suggestions for Authors

The study presented is of great interest because it includes a socio-ecological model as a theoretical reference in the study of the prediction of physical exercise. However, one of the most important problems is the lack of information on how the different variables are measured. The current status of the subject and the contributions that this study makes to what is already known today are also unclear.

The introduction is poorly informed, so a synthetic analysis of the current state of the subject should be included, both in terms of the socio-ecological model and the variables to be studied and their relationship. 

The introduction should be supported by current and relevant bibliographic citations that support the most important statements. As a general rule, statements that are not supported by previous studies are observed.

Due to the characteristics of the study, it would be interesting to establish initial hypotheses, as well as to determine more precisely the variables to be studied in the description of the objective.

Two dependent variables are included for which there is no explanation of their interest, the relationship between them in previous studies or the importance of analyzing them jointly.

Some recommendations.

Line 37. where it says “Exercise behavior” should read “Physical exercise behavior”.

Line 38. What studies support this statement? “... the effectiveness of promoting physical activity behavior among adults is suboptimal due to the multidimensional and complex nature of factors influencing their exercise participation as a result of high levels of socialization.”

Lines 41-42. Mention of “National Fitness Program” and “Healthy China 2030” should be included as a bibliographic citation and the place of consultation should be specified.

Methodology

Information on the study design should be included.

There is no information on the procedure followed in the collection of information.

Given that a reliability study of the data collection instrument is being conducted, the questionnaire should be included as supplementary material.

There is no validity study of the instrument.

The measuring instrument should be explained more precisely and clearly. It is not known how many questions there are, what the response possibilities are, or the time it takes to complete it.  The information is very scarce.

Many variables are used that are not described how they are measured. It is not clear how variables such as environment, facilities, interpersonal support are measured. Especially it is not explained how the value of the variables exercise participation and exercise time is extracted.

Results

In general, the results are repeated in text and tables. Tables should be left out and the data they contain should be removed from the text.

The stepwise logistic regression model should include the steps in which the variables are eliminated, and it is observed how the predictive capacity of the model improves. Here only the predictive value of the final model 75.68% is provided without incorporating the adjusted R-squared value in order to know what explanatory value each variable has in the model.

The following sentence should be explained more clearly and precisely “From the perspective of regression analysis, the distribution of socio-demo-graphic characteristics such as gender, age, BMI, region, marital status, work status, social status (education level, family income) of the sample was relatively reasonable.”

Lines 131 and 132 deals with contents of the methodology and not with results. It should be adjusted to this and redone.

Tables 3 and 4 should contain the t value.

Discussion

In general, the aspects that are most discussed are related to variables that have been measured in an unclear manner. As long as the measurement instruments and their relationship with other instruments used in previous studies are not clarified, it is difficult to compare data with these. The instruments define the concepts used and here it is not easy to know what we are talking about since it is not clear what each variable studied means.

The topics discussed throughout the discussion should include references to the tables of the results, as this would help the reader to better situate the topics of discussion with the results of the study. 

The results should not be repeated but should explain why and contrast them with previous studies: why are there variables that are classified as barriers, what justification is there that they may be barriers to the practice of physical activity? Why is living with parents a barrier? ...

In the discussion there is no difference between the results in the two dependent variables, exercise participation and exercise time per week.

The first topic discussed is barriers to exercise participation by analyzing predictor variables. It should be better explained why variables for which no relational data have been presented, such as BMI, are used to describe, without explaining, that this variable is interpreted as a barrier. There is no discussion of the variables noted as a barrier.

The sentence on lines 205-206 is not understood. 

One of the factors discussed in the paper are the environmental ones by contrasting large cities with small cities and explaining the benefits of the latter. This discussion is of great interest, however, this study should also analyze which factors are more determinant in the inhabitants of large cities and which factors are more determinant in small cities. 

In general, the discussion addresses issues that are of great importance and that are not sufficiently developed. An example of this is when discussing air quality. 

The issue of the role of physicians is addressed, but these data in the results provide a very limited p-value.

There are topics of interest that are not developed in the discussion. For example educational level, age, an explanation on the results of the gender variable…..

A hierarchy of the level of importance of the variables should be presented.

There is no mention of future lines of study, however there are many options that the study leaves unaddressed.

Round 2

Reviewer 2 Report

Comments and Suggestions for Authors

I would like to thank the authors of the article for their effort and patience. I consider that they have made an effort to answer the questions and suggestions raised. The work has improved considerably.

The introduction has been improved, the bibliographical references are adequately included, the variables that this study will analyze are mentioned, but there is no analysis of the current status of these variables in previous studies using the social-ecological model. Reference is also made to the difference between studies from other continents, but no arguments are given to help the reader become more aware of these differences.

The introduction could be greatly improved by including arguments explaining what we know about the study variables and what this study can contribute as a novelty to the analysis of these variables.

This analysis of the current situation would also be greatly improved by explaining what measuring instruments are used in previous studies, what problems they have and whether there is a need to create new instruments or whether instruments adapted to a different culture are really being used. 

I would like to insist, in relation to the instruments used, that the validation study that has been done is a basic and rather weak construct study, but no content validity analysis has been done. Using validated instruments is essential to know if we are really measuring what we are talking about, and if new ones are being created, this should be adequately explained.

Regarding the objective, the wording has been improved, but hypotheses in line with the results of previous studies have not been included. In this case, it should be stated in the methodology that the study is exploratory, in order to make it clear that there is no previous knowledge on the subject, if this is the case.

Regarding the methodology, I believe that the need for new measurement instruments should be better explained, along with an explanation of the inadequacy of the existing instruments and what improvement the inclusion of these new instruments would imply. 

The lack of a complete validation of the instruments should be noted as a limitation of the study.

The final instrument should be included as supplementary material. The reader does not have to discover what the instrument consists of by reading the tables, an aspect that does not allow a fluid reading of the work. On the other hand, I must insist on the need to be transparent and provide as much information as possible; the instrument is a central and basic element of any study.

Regarding the results, I must insist on compliance with international standards in the presentation of data, as well as avoiding repetition of information in text and tables. This is clearly stated in the APA guidelines.

The discussion has improved substantially, however, in some cases the results continue to be repeated without giving an explanation of these, in some cases they are contrasted with other results, but no explanations are given for these results, for example, why there are no differences between exercise behaviors and economic income levels.

Future lines of study should be included and also the factors that are a limitation in the study.

Round 3

Reviewer 2 Report

Comments and Suggestions for Authors

Dear Authors

Thank you for the effort in the responses and the changes made. However, I consider that although some of the recommendations that I have given and have not been addressed will only have an effect on the quality of the communication of the work, there are aspects that are not acceptable from my point of view.

The comments on the validity of the questionnaire, on the limitation of validity and the inclusion of the questionnaire in the actual format used should be addressed. These changes in the methodological aspects are, in my opinion, essential.
